# The Cytoprotective Role of Autophagy in Response to BRAF-Targeted Therapies

**DOI:** 10.3390/ijms241914774

**Published:** 2023-09-30

**Authors:** Ahmed M. Elshazly, David A. Gewirtz

**Affiliations:** 1Department of Pharmacology and Toxicology, Massey Cancer Center, Virginia Commonwealth University, 401 College St., Richmond, VA 23298, USA; ahmed_elshazly@pharm.kfs.edu.eg; 2Department of Pharmacology and Toxicology, Faculty of Pharmacy, Kafrelsheikh University, Kafrelsheikh 33516, Egypt

**Keywords:** vemurafenib, dabrafenib, encorafenib, autophagy, cytoprotective, cytostatic, cytotoxic

## Abstract

BRAF-targeted therapies are widely used for the treatment of melanoma patients with BRAF V600 mutations. Vemurafenib, dabrafenib as well as encorafenib have demonstrated substantial therapeutic activity; however, as is the case with other chemotherapeutic agents, the frequent development of resistance limits their efficacy. Autophagy is one tumor survival mechanism that could contribute to BRAF inhibitor resistance, and multiple studies support an association between vemurafenib-induced and dabrafenib-induced autophagy and tumor cell survival. Clinical trials have also demonstrated a potential benefit from the inclusion of autophagy inhibition as an adjuvant therapy. This review of the scientific literature relating to the role of autophagy that is induced in response to BRAF-inhibitors supports the premise that autophagy targeting or modulation could be an effective adjuvant therapy.

## 1. Introduction

This manuscript is one of a series of papers [1,2,3,4,5,6,7,8] from our research group evaluating the potential contribution(s) of the autophagic machinery to the action of various antineoplastic modalities. The overarching goal of these efforts has been to determine whether there are particular therapeutic interventions where the preclinical data, and, where available, clinical trials, support the inclusion of autophagy inhibitors as an adjuvant approach to improve therapeutic outcomes.

## 2. Overview of Autophagy

Autophagy involves the recycling of cytoplasmic components, such as the mitochondria and endoplasmic reticulum, and/or damaged organelles, for the maintenance of cellular homeostasis by the generation of energy and metabolic intermediates [3,8]. Autophagy and autophagic flux (i.e.**,** autophagy proceeding to completion) are induced in response to various stimuli such as starvation, hypoxia, oxidative stress, endoplasmic reticulum (ER) stress and protein aggregation [3,9]. A detailed description of the autophagic machinery was provided in a previous publication in this series of papers [4]. This autophagic response has also been extensively reported to be utilized by tumor cells to escape from chemotherapy-induced stress, where autophagy is serving a cytoprotective function [2]. This cytoprotective form of autophagy has been associated with the development of resistance, as, for example, in the case of tamoxifen [4]. Another form of autophagy that has received relatively limited attention is the one we have termed cytostatic, which may be associated with senescence [10]. Recently, data generated by our research group suggested the possibility that BET inhibitors in combination with tamoxifen induce a cytostatic form of autophagy in ER^+^ breast cancer [11]. Chemotherapeutic agents may also induce a cytotoxic form of autophagy under select experimental conditions and in certain experimental tumor cell lines; for instance, Zhu et al. [12] reported that irinotecan induces autophagy-dependent apoptosis in gastric cancer cells. Finally, autophagy may also play a non-protective role; for instance, our laboratory recently reported that the autophagy induced by the combination of fulvestrant plus palbociclib has this non-protective role, where no significant increase in sensitivity/cytotoxicity is evident upon combination of this standard of care treatment for ER^+^ breast cancer with autophagy inhibition [13].

## 3. BRAF-Targeted Therapies

BRAF is a serine/threonine protein kinase, encoded on chromosome 7q34 and composed of 766 amino acids [14,15]. BRAF has the highest capacity among the other RAF family proteins, ARAF and CRAF, to phosphorylate MEK (MEK1 and MEK2) [16]. MEK then activates ERK (ERK1 and ERK2) in the cytoplasm, leading to ERK translocation to the nucleus where it phosphorylates various transcription factors. The Ras-Raf-MEK-ERK (also known as MAPK/ERK) pathway regulates several critical cellular processes that include proliferation, differentiation as well as apoptosis [14,15]. Ras-Raf-MEK-ERK dysregulation has been reported to contribute to the development of several tumor types due to mutations in constituent proteins of this pathway, including RAS (KRAS and NRAS) and RAF (BRAF) [17,18]. BRAF mutations are frequently acquired in various malignancies including thyroid carcinoma, colorectal cancer and especially melanoma [14,15,19]. Mutations that have been identified include the following: BRAFV600K, substituting lysine for valine GTG > AAG; BRAFV600R, GTG > AGG; BRAF V600′E2′, GTG > GAA; BRAF V600D, GTG > GAT; as well as the most frequent mutation BRAFV600E, GTG > GAG, where glutamic acid is substituted for valine [14,20].

BRAF-targeted therapeutics that have been developed and are being utilized in the clinical setting include vemurafenib, dabrafenib as well as encorafenib. Dabrafenib and encorafenib have recently been utilized in combination with the MEK inhibitors trametinib [21] and binimetinib [22], respectively. Despite the initial clinical successes with these agents, as is often the case with other chemotherapeutic modalities, resistance development constrains their clinical effectiveness. While different molecular mechanisms of resistance have been identified [23], this article examines the literature relating to autophagy, and provides an overview on the relationship between autophagy and BRAF-inhibitors in various tumor models. The goal of this effort, as in our previous work, is to determine whether autophagy targeting or modulation could be an effective strategy for enhancing BRAF-targeted therapeutic activity.

## 4. Vemurafenib and Autophagy

Vemurafenib is a selective inhibitor for mutated BRAF V600E kinase, reducing signaling through the MAPK pathway [24]. Vemurafenib was approved in 2011 for the treatment of patients with unrespectable or metastatic melanoma harboring BRAFV600E mutations [25]. Although vemurafenib has demonstrated clinical efficacy, most, if not all, patients develop resistance, and ultimately disease progression, at a median time of approximately 6 months [26]. Among the different molecular mechanisms that contribute to vemurafenib resistance, several studies have implicated the autophagic machinery. For example, Ma et al. [27] published in-depth studies evaluating the relationship between autophagy and BRAF inhibition in melanoma. Initial studies evaluated whether vemurafenib/dabrafenib induced the autophagic machinery in patient samples with BRAF-mutant melanoma by staining for the autophagy marker, LC3II. The patients’ samples were evaluated prior to treatment (pretreatment samples) and upon re-growth, despite treatment with vemurafenib/dabrafenib (resistance samples) [27]. LC3II scores in pretreatment samples were found to be lower than in resistance samples in 74% of cases, equivalent in 13% of cases and higher in 13% of cases [27]. Analysis by electron microscopy of several tumor biopsies from two BRAF-mutant melanoma patients enrolled in clinical trials for vemurafenib indicated a two- to six fold increase in autophagic vacuoles compared with baseline measurements. Additionally, a biopsy at the time of tumor progression revealed persistently elevated levels of autophagy in both patients [27]. These data suggested that autophagy is induced in response to BRAF inhibitors and remains activated at the time of tumor progression. The elevated autophagy in response to vemurafenib/dabrafenib in the resistance samples may reflect a survival role for autophagy in this disease upon treatment with BRAF inhibitors.

Using a panel of melanoma cell lines (BRAF inhibitor-sensitive (A375P, SKMEL5, MEL1617) or BRAF inhibitor-resistant (MEL1617R, WM983BR, MEL624)), these investigators studied the possible relationship between autophagy and vemurafenib both in vitro and in vivo [27]. Interestingly, vemurafenib treatment resulted in autophagy induction in all cell lines, as evidenced by the significant elevation in the LC3II/LC3I ratio and decline in p62/SQSM1 levels [28]. Autophagy induction was further confirmed using the mCherry-eGFP-LC3 assay in A375P cells, where vemurafenib induced the formation of small red puncta, similar to the autophagy inducer, rapamycin, reflecting increased autophagosome production and ongoing autophagic flux.

Ma et al. [27] further performed BRAF knockdown studies using shRNA in A375P cells; here, autophagy induction was shown by an elevated LC3II/LC3I ratio, indicating that BRAF inhibition is associated with autophagy induction. Importantly, BRAF deletion in combination with the pharmacological autophagy inhibitor, HCQ, produced a threefold increase in growth inhibition compared to the control, reflecting the cytoprotective role of the autophagy induced by BRAF inhibition. The cytoprotective role was further confirmed genetically using ATG5-directed shRNA, where autophagy inhibition in combination with vemurafenib heightened growth inhibition as compared to vemurafenib alone in the resistant MEL624 cells. Although enhanced growth inhibition was not evident in the sensitive A375P cells, autophagy inhibition was associated with increased suppression of colony formation, as assessed by a clonogenic survival assay. The combination of autophagy inhibition together with vemurafenib als resulted in enhanced suppression of clonogenicity in MEL1617R and MEL624 cells. In further support of the cytoprotective function of autophagy, HCQ in combination with vemurafenib promoted increased cell death in BRAF inhibitor-sensitive A375P, WM983B and MEL1617 cells as well as in BRAF inhibitor-resistant MEL624, MEL1617R and WM983BR cells grown as 3D spheroids [27]. Similar results were also obtained with another autophagy inhibitor, Lys05, which resulted in additive cytotoxicity upon combination with vemurafenib in A375P, SKMEL5, MEL1617 and MEL624 cells.

These investigators then tested the combination of autophagy inhibition together with vemurafenib in vivo using xenograft mice models injected with BRAF inhibitor-resistant MEL624 cells [27]. Vemurafenib in combination with Lys05 showed significant tumor regression as compared to each drug alone. Vemurafenib alone promoted significant p62/SQSM1 degradation and an elevated LC3II/I ratio, as well as increased autophagic vacuole formation by electron microscopy, confirming that vemurafenib induces autophagy in this model. Conversely, the combination of vemurafenib and Lys05 resulted in p62/SQSM1 accumulation, evidence of autophagic blockade, accompanied by increased cleaved-caspase 3, indicative of apoptosis [27]. These results markedly reflect the cytoprotective role of vemurafenib-induced autophagy in these experimental model systems, highlighting the possibility that autophagy targeting could be a possible strategy for enhancing tumor cell responsiveness to vemurafenib.

The ER stress response has been associated with the autophagic machinery and can be activated via stimulation of the ER chaperone, GRP78, which binds to and limits activation of three transmembrane proteins: inositol-requiring kinase 1α (IRE1α), activating transcription factor 6α (ATF6α) and PKR-like ER-kinase (PERK) [7,27,29,30,31]. The PERK arm of the ER stress response is closely associated with the autophagic process [32,33]. The activation of PERK leads to the phosphorylation of eukaryotic initiation factor 2α (eIF2α), together with increasing the expression of the transcription factor activating transcription factor 4 (ATF4) as well as C/EBP homologous protein (CHOP) [27]. Ma et al. [27] investigated the potential mechanism(s) whereby vemurafenib induces autophagy. Initially, these investigators showed that vemurafenib treatment induced an ER stress response based on ER stress-associated morphological changes such as ER dilation and disorganization with granular contents, suggestive of retained unfolded proteins. Additionally, phospho-PERK, ATF4, phospho-eIF2α and CHOP levels were upregulated upon vemurafenib treatment in both BRAF inhibitor-sensitive A375P and BRAF inhibitor-resistant MEL624 cells. Vemurafenib also induced IRE1α signaling. The ER stress response was further confirmed by the binding between mutant BRAF and GRP78 upon vemurafenib treatment, and the dissociation of GRP78 from PERK. To further validate the critical role of PERK in vemurafenib activity, these investigators demonstrated that upon combining vemurafenib with the PERK inhibitor, GSK2606414, cleaved PARP and cleaved caspase-3 were increased in MEL624 and A375P cells together with a reduction in the viability of MEL624 cells. PERK inhibition entirely abrogated the vemurafenib-induced autophagy, which was further confirmed by depleting PERK using siRNA. These results further confirm the cytoprotective role of vemurafenib-induced autophagy and the critical underlying role played by PERK in the autophagy induction.

In addition to the epigenetic readers/regulators which may control the fate of the autophagic flux, as discussed in previous publications [8,34], several studies investigated the potential control of the autophagic machinery by microRNAs (miRNAs). miRNAs are non-coding RNAs that bind the 3′-untranslated region (3′-UTR) of target mRNAs, inhibiting their translation or causing their degradation, which ultimately results in the suppression of gene expression [35]. Several studies investigated the association between miRNA and the autophagic pathways [36]; for instance, Shan et al. [37] reported that miR-17-5p, miR-30a, miR-216, miR-376 as well as miR-409-3p inhibit Beclin1 expression, ultimately suppressing autophagy in different tumor models.

Luo et al. [38] studied the relationship between miR-216b, autophagy and vemurafenib in melanoma cells harboring a V600E mutation in the BRAF gene; vemurafenib-sensitive A375 and G-361 cell lines, as well as vemurafenib-resistant G-361-R and A375-R cells, were utilized in these studies. Consistent with their differencing sensitivities, vemurafenib treatment resulted in the inhibition of ERK phosphorylation in the sensitive but not in the resistant cell lines [38]. The level of miR-216b was downregulated in both sensitive and resistant cell lines upon vemurafenib treatment in a dose-dependent manner. The level of miR-216b in MeWo melanoma cells carrying the wild-type BRAF gene was not significantly altered by vemurafenib, suggesting an association between miR-216b downregulation upon vemurafenib treatment and BRAF mutations [38].

With regard to autophagy, and consistent with the observations by Ma et al. [27], vemurafenib treatment induced the autophagic machinery in A375 melanoma cells, as shown by GFP-LC3 puncta formation [38]. The induction of autophagic flux was further confirmed using western blotting; specifically, LC3I/II conversion and p62/SQSM1 degradation were evident. Interestingly, miR-216b overexpression suppressed both the basal level as well as the vemurafenib-induced autophagy. Conversely, autophagic flux significantly increased upon miR-216b inhibition, consistent with miR-216b acting as an autophagy suppressor [38].

Upon using bioinformatics tools for the screening of the different autophagic genes for possible binding sites of miR-216b, Beclin1, UVRAG and ATG5 were identified [38]. This relationship was validated in studies where miR-216b overexpression suppressed the mRNA and protein levels of Beclin1, UVRAG and ATG5 in A375 cells; consistently, anti-miR-216b transfection upregulated the mRNA and protein levels of these three genes, confirming that miR-216b has a negative regulatory role for the autophagic machinery [38]. It was further shown that miR-26b overexpression, which suppresses autophagy, was able to enhance the response to vemurafenib of both the vemurafenib-sensitive A375 and G-361 cells, and the drug-resistant G-361-R and A375-R cells [38]. In contrast, miR-216b inhibition suppressed vemurafenib-induced activity; furthermore, miR-26b knockdown markedly increased the clonogenic growth of A375 cells after vemurafenib treatment, all of which supports a cytoprotective role for the autophagy induced by vemurafenib.

Luo et al. [38] further validated these results in vivo using xenograft tumors in nude mice models injected with sensitive and resistant melanoma cell lines. They reported that miR-216b overexpression in combination with vemurafenib showed greater antitumor activity compared with vemurafenib alone [38]. These investigators confirmed that the combination’s enhanced antitumor activity is through autophagy inhibition, as shown by the reduction in the LC3II/I ratio as well as the attenuated p62/SQSM1 degradation in both sensitive and resistant tumors [38], confirming the cytoprotective role of the autophagic machinery induced by vemurafenib in this model.

Goodall et al. [39] showed that vemurafenib effectively blocked oncogenic BRAF signaling, as shown by reduced phosphorylation of the downstream effector, ERK1/2, with no effect on the autophagic flux, as indicated by unchanged levels of LC3II in A375 melanoma cells. This lack of effect appears to be in direct contrast to the findings of Ma et al. [27] and Luo et al. [38]. However, the A375 cells were shown to have high levels of basal autophagy, as determined by the significant LC3II accumulation upon CQ addition. The combination of vemurafenib with different autophagy inhibitors, CQ, QN [39], VATG-027 [39,40] as well as VATG-032 [39,40]**,** in A375 cells reduced colony formation as compared to vemurafenib alone. Similar results were generated upon combining the autophagy inhibitors with the catalytic mTOR inhibitor, AZD8055 [39,41]**,** that promotes autophagy. These results may reflect the fact that autophagy is playing a role in A375 cell survival whereas vemurafenib did not induce autophagic flux in BRAF V600E-mutated melanoma cells. What is missing for a more complete picture are experiments incorporating the genetic inhibition of autophagy as well as multiple autophagy markers to validate the cytoprotective role of autophagy in this cell line [28].

Moving away from studies limited to melanoma cells, Wang et al. [31] studied the possible targeting of autophagy to increase the effectiveness of vemurafenib treatment using the anaplastic thyroid carcinoma BRAF-mutated FRO cell line as well as papillary thyroid carcinoma BRAF-mutated BCPAP cells. Both cell lines, BCPAP and FRO cells, were relatively resistant to vemurafenib, with IC_50_ values of 900 nM and 6000 nM, respectively. Interestingly, vemurafenib treatment increased the ratio of LC3II/LC3I in both cell lines as well as in a third thyroid cancer cell line, 8505C cells, in a dose- and time-dependent manner, indicative of autophagy induction. These results were further confirmed using a GFP-LC3 assay in FRO cells, where vemurafenib treatment increased the puncta number, indicating increased autophagosome formation, as detected by a transition electron microscope (TEM). The role of the induced autophagy was interrogated by combining vemurafenib with HCQ in both BCPAP and FRO cells, where the combination showed a greater reduction in cell growth and clonogenic capacity than each drug alone. These experiments support the contention that the form of autophagy induced by vemurafenib treatment is also cytoprotective in thyroid carcinoma cell lines. The cytoprotective form of autophagy was further confirmed using genetic inhibition approaches, specifically in that ATG5 depletion via siRNA in combination with vemurafenib effectively reduced cell viability to a greater extent than vemurafenib alone. The potential therapeutic benefits of combining autophagy with vemurafenib in vivo was interrogated further using xenograft mice tumors models injected with BCPAP cells. The accumulation of autophagic vacuoles in response to vemurafenib was detected by electron microscopy. Furthermore, the combination of vemurafenib and HCQ showed a greater reduction in tumor growth compared to each drug alone, with no signs of toxicity.

As mentioned earlier, once the ER capacity for degrading unfolded/misfolded proteins becomes saturated, the ER stress response can be activated via stimulation of the ER chaperone, GRP78, and three signaling pathways: PERK/eIF2α/CHOP, IRE-1/Xbp-1 and ATF6α. Upon this occurrence, alternative pathways for the degradation of these proteins are induced [7,30,31]**.** Vemurafenib treatment induced the phosphorylation of eIF2α and CHOP proteins, suggesting that vemurafenib treatment promotes an ER stress response which may be related to the induced autophagic flux. The association between the ER stress response and the autophagic flux induced by vemurafenib was further confirmed using the PERK inhibitor, GSK2606414, where the ratio of LC3II/LC3I was reduced, together with suppression of p62/SQSM1 digestion and reduced CHOP levels, suggesting that the ER stress response plays a crucial role in inducing autophagy during vemurafenib treatment in thyroid cancer; these findings mirror those of Ma et al. [27] in the melanoma model.

Consistent with the findings of Wang et al. [31], Run et al. [42] showed that vemurafenib treatment induced autophagy in thyroid cancer cells based on increases in Beclin1 and the LCII/I ratio as well as p62/SQSM1 degradation. Furthermore, pharmacological autophagy inhibition using 3-MA in combination with vemurafenib significantly reduced cell viability of the vemurafenib-resistant BCPAP-R cells, together with increasing apoptosis, suggestive of a cytoprotective role of autophagy in this model. Notably, they showed that HMGB1 targeting reversed the vemurafenib-induced increase in Beclin1 expression, increased LC3II and decreased p62/SQSM1 protein levels, suggesting a possible role of HMGB1 in vemurafenib-induced cytoprotective autophagy.

Using a colorectal cancer model, Hu et al. [43] studied the natural alkaloid lycorine and its potential utilization in combination with vemurafenib. By monitoring cell viability and apoptosis (annexin V/PI staining), they showed that lycorine alone induces a dose-dependent reduction in the viability of HCT116, SW480, RKO and CT26 colorectal cancer cell lines together with inducing a late stage of apoptosis. Importantly, they showed that lycorine promoted the autophagic machinery, as shown by increased autophagosome numbers in HCT116 cells detected by TEM, and the LC3-GFP-RFP assay in SW480 and HCT116 cells, as well as increasing Beclin1 and LC3II levels by western blotting. Interestingly, the autophagy induction was accompanied by a reduction in the mitochondrial membrane potential, as well as a significant increase in the Bax/Bcl-2 ratio, raising the possibility that apoptosis and autophagy induction may occur in parallel with lycorine treatment.

Subsequent experiments indicated that lycorine may interact with MEK2, as shown by CDOCKER docking; this was confirmed by downregulation of MEK2 phosphorylation and the p-ERK/ERK ratio, the downstream effectors of MEK2. Overexpression of MEK2 in HCT116 cells treated with lycorine, whereupon lycorine failed to induce autophagy or apoptosis, appeared to confirm that MEK2 is the primary mediator for lycorine effects in colorectal cancer cell lines. Conversely, MEK2 deletion in MEK2-overexpressing cells, via targeting with shRNA, restored the levels of apoptosis as well as autophagy after lycorine treatment in HCT116 cells. As BRAF may be persistently activated via gain-of-function mutations, and these constitutively activating signals pass to ERK1/2 through MEK1/2 [44], these investigators combined lycorine with vemurafenib in HCT116 cells, where the combination showed an enhanced reduction in cell viability compared to each drug alone. Interestingly, they reported that the combination has a more significant effect on MEK2-overexpressing cells than lycorine alone, as shown by annexin V/PI flow cytometry. The combination also showed similar results in the SW480 cell line.

Further studies of the effect of lycorine in combination with vemurafenib were performed in vivo using xenograft nude mouse models. Here, the combination showed a significantly greater antitumor activity than each drug alone, together with increasing the BAX/BCL2 ratio and LC3II levels, as shown by immunohistochemistry analysis. These results appear to suggest that lycorine in combination with vemurafenib increases the extent of autophagic flux to a greater extent than lycorine alone, and may be reflective of a cytotoxic role of autophagy because of its association with apoptosis induction; however, here it is necessary to emphasize that the autophagy induced primarily by lycorine appears to be cytotoxic and that neither vemurafenib-induced autophagy nor the autophagy induced by the interaction between these two agents were investigated. Furthermore, both pharmacological and genetic inhibition of autophagy would be required needed to confirm the role of the induced autophagy in this model [28].

Collectively, with a very few minor reservations, the majority of the studies presented strongly support the premise that vemurafenib induces autophagy in different tumor models, that vemurafenib-induced autophagy is mediated through ER stress/PERK [27,31], and that the cytoprotective role of autophagy predominates, suggesting the potential clinical utility of targeting the autophagic pathway to increase the effectiveness of vemurafenib in different tumors.

## 5. Dabrafenib and Autophagy

Dabrafenib is a potent inhibitor for mutated BRAF that is currently being utilized as the first or subsequent line treatment for patients with unresectable or metastatic melanoma with BRAF V600 mutations [45]. Dabrafenib has shown a greater selectivity for BRAF with less potency for CRAF than vemurafenib [46]. Furthermore, dabrafenib in combination with trametinib, a MEK inhibitor [21], demonstrated a higher response rate and improved clinical efficacy [45]; consequently, the FDA approved these two drugs as monotherapies as well as in combination for treating patients with BRAF-mutant metastatic melanoma [47].

Diverse mechanisms of resistance to dabrafenib have been considered [23], with a recent focus on the involvement of autophagy. In addition to the findings of Ma et al. [27] (mentioned earlier), which showed that dabrafenib treatment is associated with autophagy induction in patient samples with melanoma, Yu et al. [48] studied the relationship between dabrafenib and autophagy using A375 and MEL624 melanoma cell lines. These investigators showed that dabrafenib induced autophagic flux in both cell lines, as indicated by GFP-LC3 puncta formation, where dense LC3-positive puncta were observed upon dabrafenib treatment. Autophagy induction was further validated using western blotting, with LC3I/II conversion and p62/SQSM1 degradation, in a dose-dependent manner. Treatment of dabrafenib-treated cells with either of the pharmacologic autophagy inhibitors, 3-MA or CQ, reduced the viability of both A375 and MEL624 melanoma cell lines as compared to dabrafenib alone, supporting a cytoprotective role for autophagy. However, genetic inhibition studies are needed to further validate the protective role of autophagy in this experimental model [28].

Mechanistically, dabrafenib treatment was shown to result in the upregulation and downregulation of HMGB1 and miR-26a, respectively, in A375 and MEL624 melanoma cell lines. Importantly, miR-26a overexpression significantly suppressed dabrafenib-induced autophagy, as shown by GFP-LC3 puncta formation, together with a robust elevation in cleaved PARP and a reduction in HMGB1 levels. These results suggested that miR-26a has a negative regulatory effect on the autophagic flux induced by dabrafenib, similar to miR-216b suppressing the effect on vemurafenib-induced autophagy, as mentioned by Luo et al. [38]. Additionally, they showed that elevated levels of HMGB1 are required for dabrafenib-induced cytoprotective autophagy, as shRNA-mediated HMGB1 knockdown is accompanied by a reduction in LC3II levels as well as GFP-LC3 puncta formation. Furthermore, HMGB1 deficiency caused a significant elevation in cleaved PARP levels. The accompanied overexpression of miR-26a, together with HMGB1 depletion, caused the cells to be more sensitive to the dabrafenib treatment as compared to the controls, highlighting that HMGB1 is needed for inducing cytoprotective autophagy in response to dabrafenib, which is antagonized by miR-26a.

Awada et al. [49] performed a clinical trial for dabrafenib and the MEK inhibitor, trametinib [50], in combination with HCQ, in advanced BRAFV600-mutant melanoma patients previously treated with BRAF-/MEK-inhibitors and immune checkpoint inhibitors. Patients were randomized between upfront treatment with dabrafenib, trametinib and HCQ (experimental arm), or dabrafenib and trametinib, with the potential to add HCQ once the tumor progressed (contemporary control arm). They reported that dabrafenib and trametinib showed no new safety signals, while HCQ was suspected of promoting the development of an anxiety/psychotic disorder in one patient. The objective response as well as the disease control rates were 20.0% and 50.0%, respectively, in the experimental arm, whereas in the contemporary control arm no responses were observed before or after HCQ addition, with a negative evaluation of the risk/benefit ratio for adding HCQ to dabrafenib and trametinib. Similarly, Mehnert et al. [51] recently performed a Phase I/II trial of the dabrafenib and trametinib in combination with HCQ in advanced BRAFV600-mutant melanoma (BAMM trial). In this study, the combination was well-tolerated and showed a strikingly high response rate; however, while the combination showed a transient response in many cases, unfortunately the pre-specified progression-free survival was not reached. Overall, the evidence for clinical activity was observed with a complete response rate of 41% across the entire cohort; additionally, the combination therapy appeared promising in patients with elevated LDH with prior treatment, with a response rate of 88%.

These results reflect that autophagy may play a cytoprotective role in response to dabrafenib; the clinical data, while relatively limited, nevertheless appear to support autophagy targeting as a potentially beneficial strategy to increase dabrafenib and trametinib effectiveness.

## 6. Encorafenib and Autophagy

Encorafenib targets BRAF V600E-, V600D- and V600K-mutant kinases [52]. Recently, encorafenib in combination with binimetinib, a MEK inhibitor [22], was approved by the FDA for the treatment of patients with unresectable or metastatic melanoma with BRAF V600E or V600K mutations, based on findings from the COLUMBUS trial [52]. To investigate the potential involvement of autophagy in modulating the response to encorafenib, Li et al. [53] studied encorafenib in four human melanoma cell lines harboring the BRAFV600E mutation, A375, G361, RPMI7951 and SK-MEL-24 cells, while using C8161, a human melanoma cell line carrying wild-type BRAF/NRAS, as a control. Encorafenib exhibited a potent antiproliferative effect, suppressing colony formation and driving the A375, G361 and SK-MEL-24 cells into G1 arrest, but not the RPMI7951 and C8161 cells; however, no apoptosis was reported, as detected by annexin V and PI /flow cytometry. Encorafenib also caused a reduction in cell cycle regulatory proteins, Cyclin 1 and CDC6, in both A375 and G361 cells, together with suppressing pERK in A375, G361 and SK-MEL-24 cells, but not the RPMI7951 and C8161 cells. Interestingly, encorafenib induced a senescence phenotype in in A375, G361 and SK-MEL-24 cells in a time-dependent manner, as shown by increased β-galactosidase activity, together with a reduction in the percentage of BrdU-positive cells. Furthermore, encorafenib treatment in A375 and G361 cells resulted in increased levels of p27^KIP1^, a reduction in Rb phosphorylation as well as an upregulation of p21^CIP1^ in A375 but not G361 cells, suggesting that encorafenib-induced senescence may be mediated via p27^KIP1^ as well as Rb. In addition to senescence induction, encorafenib also induced autophagy in A375 and G361 cells, as detected by the autophagosome formation observed by TEM, a reduction in p62/SQSTM1 levels as well as a reduction in LC3II levels, the latter suggesting active autophagic flux; the latter was confirmed using the GFP-mRFP-LC3 assay, where encorafenib treatment increased the red LC3-positive vacuoles, representing autolysosomes, in both A375 and G361 cells. Mechanistically, encorafenib treatment resulted in a dose-dependent reduction in the phosphorylation state of mTOR, together with its downstream effector, p70S6K, in A375 and G361 cells, consistent with the promotion of autophagy.

Li et al. [53]**,** in studying the relationship between senescence and autophagy in this system and whether these responses occur in parallel, showed that autophagy inhibition pharmacologically, using BAF A1, or genetically, using LC3-targted shRNA, decreased β-galactosidase activity, together with attenuating the encorafenib-induced p27^KIP1^ expression and Rb activation, suggesting that senescence induction is dependent on the autophagic flux. As RPMI7951 cells were resistant to the encorafenib, studies were performed to modulate autophagy in combination with encorafenib using the autophagy inducers rapamycin and BEZ235. These combinations yielded significant growth-inhibiting effects compared with each treatment alone. Similar results were obtained in A375 cells, where the combination increased the extent of the growth inhibition [10,54]. Here, a cytostatic function of encorafenib-induced autophagy is implicated.

Furthermore, Hartman et al. [55] studied encorafenib efficacy in the patient-derived melanoma cell lines, DMBC11, DMBC12, DMBC21, DMBC28 and DMBC29, harboring the BRAFV600E mutation. Initially, encorafenib treatment was shown to reduce the viability of all the cell lines tested. The reduction in cell viability was accompanied by autophagy induction, as evidenced by LC3 II accumulation and p62/SQSM1 degradation. Furthermore, senescence was also apparently induced in all the cell lines, with the exception of the DMBC29 cells, as shown by increased β-galactosidase activity. It was also found that encorafenib treatment reduced the levels of c-Myc and CCND1 mRNA and the protein levels of ERK1/2. These results are indicative of an active autophagic flux together with the triggering of senescence in melanoma cell lines, which is consistent with the findings of Li et al. [53]. Interestingly, they showed that encorafenib only induced apoptosis in DMBC21 and DMBC28 cells, based on elevated caspase-3/7-positive cells as well as cleaved PARP. Additionally, the targeting of MCL-1 by S63845 [56] was able to sensitize the melanoma cell lines to encorafenib. These findings by Hartman et al. [55] indicate that encorafenib induced the autophagic machinery together with senescence; however, the role of the autophagy in this system, i.e.**,** whether it is cytostatic or cytoprotective, was not defined.

## 7. Conclusions

BRAF inhibitors have demonstrated a substantial degree of efficacy in the treatment of patients with melanoma carrying BRAF V600 mutations [45]; however, as is the case with other chemotherapeutic agents, resistance development constrains the clinical utility of these drugs. As mentioned previously [3,4,7], autophagy is a survival mechanism, with four different roles that have been identified in response to different antineoplastic modalities; cytotoxic, cytostatic, non-protective and, the resistance-associated form, cytoprotective [5]. It is important to highlight that the nature of the autophagy being induced is generally dependent on both the chemical nature of the compound as well as the cell line/tumor model being utilized. Different clinical trials have recently emerged in efforts to increase the effectiveness of various chemotherapeutic agents by targeting the autophagic machinery, using pharmacological autophagy inhibitors HCQ and CQ [3,7]. Furthermore, recent studies have investigated the possibility of modulating the unregulated autophagic pathway [57,58]. As summarized in Table 1, the majority of studies for vemurafenib strongly support the premise that the cytoprotective role of autophagy is being induced as well as that a linkage is established between the ER stress response and the vemurafenib-induced autophagy in different tumor models. These findings underlie the possibility of autophagy targeting and its possible clinical translation to increase vemurafenib efficacy. Similarly, dabrafenib showed induction of the cytoprotective form of autophagy. Moreover, the clinical data suggest the potential for successfully combining autophagy inhibition with dabrafenib and trametinib. Regarding the newly approved BRAF inhibitor, encorafenib, only limited information is available in the scientific literature relating to this agent and autophagy. Nevertheless, BRAF inhibition appears to hold promise as a therapeutic modality where autophagy inhibition could potentially serve to sensitize malignancies and possibly prevent or delay the development of drug resistance.

## Figures and Tables

**Table 1 ijms-24-14774-t001:** The role of the autophagic machinery in response to BRAF inhibitors.

BRAF Inhibitor	Cell Lines/Tumor Type	Autophagy Induced/Suppressed	Role of Autophagy	Reference
Vurmuarfenib	Patients’ samples, melanoma cell lines, BRAF inhibitor-sensitive (A375P, SKMEL5, MEL1617) or BRAF inhibitor-resistant (MEL1617R, WM983BR, MEL624), 3D culture models, as well as in vivo using xenograft mice models injected with BRAF inhibitor-resistant MEL624 cells	Autophagy induced	Cytoprotective	[27]
Vemurafenib	Vemurafenib-sensitive; A375 and G-361 cell lines, vemurafenib-resistant; G-361-R and A375-R cells, as well as xenograft tumors in nude mice models injected with sensitive and resistant melanoma cell lines	Autophagy induced	Cytoprotective	[38]
Vemurafenib	A375 melanoma cells	Autophagy not induced	NA	[39]
Vemurafenib	The anaplastic thyroid carcinoma BRAF-mutated FRO cell line, papillary thyroid carcinoma BRAF- mutated BCPAP cells as well as in vivo using xenograft mice tumor models injected with BCPAP cells	Autophagy induced	Cytoprotective	[31]
Vemurafenib	Thyroid cancer cells and vemurafenib-resistant BCPAP-R cells	Autophagy induced	Cytoprotective	[42]
Vemurafenib	HCT116, SW480, RKO and CT26 colorectal cancer cell lines, as well as in vivo using xenograft tumors in nude mouse models	Autophagy induced in response to lycorine and vurmurafenib	Cytotoxic mainly in response to lycorine	[43]
Dabrafenib	Patients’ samples with melanoma	Autophagy induced	Cytoprotective	[27]
Dabrafenib	A375 and MEL624 melanoma cell lines	Autophagy induced	Cytoprotective	[48]
Dabrafenib	Clinical trial in patients with advanced BRAFV600-mutant melanoma	NA	NA	[49]
Dabrafenib	Clinical trial in patients with advanced BRAFV600-mutant melanoma	NA	NA	[51]
Encorafenib	Human melanoma cell lines harboring the BRAFV600E mutation; A375, G361, RPMI7951 and SK-MEL-24 cells, while using C8161, a human melanoma cell line carrying wild-type BRAF/NRAS, as a control	Autophagy and senescence induced	Cytostatic	[53]
Encorafenib	Patient-derived melanoma cell lines: DMBC11, DMBC12, DMBC21, DMBC28 and DMBC29	Autophagy and senescence induced	Cytostatic or cytoprotective	[55]

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
