# Peer review of "The Cytoprotective Role of Autophagy in Response to BRAF-Targeted Therapies"

_ijms, 2023, doi:10.3390/ijms241914774_

Round 1
Reviewer 1 Report
A review by Elshazly and Gewirtz discusses a contribution of autophagy to response to therapies targeting mutated BRAF. While the subject is interesting and of clinical relevance, the manuscript needs several improvements.
Specific comments:
1. Introduction - while the contribution of the Authors to the field of autophagy is important and, I would recommend to refer to own papers within the text and in an appropriate contexts, not only by mentioning the contribution. The introduction written as it is is not necessary at all. Please modify or remove.
2. Table 1 - correct "Vurmuarfenib" to "vemurafenib".
3. A study assessing changes in autophagy in response to encorafenib is missing: doi: 10.1016/j.canlet.2020.11.036
4. The Authors discusses three inhbitors of BRAFV600 separately. Are there any differences in autophagy-related responses when these three inhibitors are compared. This issue might be discussed and could substantially improve the manuscript.
5. Please correct typos.
Reviewer 2 Report
In this review “The cytoprotective role of autophagy in response to BRAF-targeted therapies” the authors have provided an in-depth role of autophagy in triggering resistance to BRAF inhibitors. The authors have provided a comprehensive review of published articles showing the autophagy-mediated resistance for chemotherapeutic drugs currently approved by the FDA. Additionally, this article highlights the importance of targeting autophagy machinery using specific inhibitors as combination therapy alongside BRAF inhibitors. This article is scientifically sound and should be published by IJMS. Below are some comments that can further increase the impact of this manuscript:
1. Authors have cited themselves quite often in the text; though it is not inappropriate, I think the language should be improvised to make it not look like the authors are providing a report on their previously published work.
2. Authors should consider providing a graphic figure describing the autophagy mechanism.
3. Authors should describe how autophagy is triggered and leads to resistance to current chemotherapeutic agents.
4. Authors have mentioned specific experiments from the articles they have reviewed; it is difficult to understand without showing a figure or scheme for those experiments and their conclusions. Authors should consider improvising their review by providing a figure to support their decision or modifying the text to make it more impactful.
